# Enhancing a 3D Foundation Model with Gaussian Sampling for Interactive Biomedical Image Segmentation

Jianhang Ji[1][0000−0002−7733−3343], Tao Han[1,2,∗][0000−0002−5271−0528], Tingyi Lin[1][0009−0009−5117−1700], and Junchen Xiong[1][0009−0000−1988−1184]

[1] The Hanglok-Tech Company, Ltd., Hengqin, China
[2] The Department of Radiology, Zhongda Hospital, Medical School, Center of Interventional Radiology & Vascular Surgery, Southeast University, Nanjing, China
107000292@seu.edu.cn

**Abstract.** Interactive segmentation of 3D medical images seeks to produce accurate object masks with minimal user input, substantially alleviating the burden of manual annotation. For the CVPR 2025 Foundation Models for Interactive 3D Biomedical Image Segmentation Challenge, we extend the VISTA3D foundation model—a state-of-the-art 3D segmentation network supporting both automatic and interactive modes—by introducing several targeted improvements for robust interactive segmentation. First, we propose a Gaussian Edge-Center point sampling strategy, which leverages Gaussian-weighted randomness combined with center/edge distance transforms to preferentially sample points at object centers and boundaries. This yields more realistic and effective foreground/background click simulations during training. Second, we integrate this sampler into a two-stage fine-tuning pipeline: initial conventional fine-tuning with provided pre-trained weights, followed by prompt-focused fine-tuning using our improved sampling strategy. Third, to meet the challenge's 90-second runtime limit, we optimize inference by dynamically adjusting the region of interest (ROI) size and resolution based on input voxel spacing, including adaptive downsampling and ROI cropping. We trained models for both tracks—using 4×A100 GPUs for the full dataset and 4×A800 GPUs for the 10% core dataset—under identical protocols. On the validation set, our full-data model achieved a Dice Similarity Coefficient (DSC) Final of 0.7194, while the core-data model achieved 0.6782. These results demonstrate that our enhanced approach effectively leverages the capabilities of foundation models for interactive 3D segmentation, delivering accurate results with efficient user interaction. Code is available at https://github.com/M4cheal/GS_MedSegFM.

**Keywords:** Interactive segmentation · 3D biomedical image analysis · Prompt sampling

## 1 Introduction

3D biomedical image segmentation is essential for clinical diagnosis, surgical planning, and biomedical research [15,4,8]. The advent of high-resolution imag-

ing modalities such as CT, MRI, PET, and ultrasound has created an urgent need for segmentation models that are accurate, robust, and broadly applicable to diverse anatomical structures and imaging protocols [13,16]. However, manual annotation of 3D medical images is extremely labor-intensive and time-consuming, especially for complex structures or rare pathologies [1,9]. This challenge has motivated the development of interactive segmentation algorithms, where the model iteratively refines its predictions based on sparse user prompts (such as points or bounding boxes), thereby minimizing the annotation burden while maintaining high accuracy [7,9,12].

The CVPR 2025 Foundation Models for Interactive 3D Biomedical Image Segmentation Challenge further raises the bar by requiring universal models capable of segmenting a wide variety of anatomical structures across multiple imaging modalities, with efficient and accurate human-in-the-loop refinement. This goal is complicated by heterogeneous data, anatomical variability, class imbalance, and strict computational constraints such as a 90-second runtime limit per case.

Recent years have witnessed rapid progress in foundation models for both natural and medical image segmentation. The Segment Anything Model (SAM) [7] and its successors [14] have pioneered promptable segmentation, enabling user-driven object extraction via points, boxes, or text prompts. Inspired by these advances, the medical imaging community has developed powerful 3D segmentation foundation models such as MedSAM [9], MedSAM2 [11], SegVol [2], SAM-Med3D [17], and nnInteractive [5]. These approaches extend promptable segmentation to 3D medical images, leverage large and diverse datasets, and support interactive correction, but often still face limitations: many methods rely on 2D interaction strategies, require slice-by-slice annotation, or struggle with efficient and robust refinement in volumetric settings. VISTA3D [3] addresses several of these challenges with a unified 3D foundation model supporting both automatic and interactive segmentation. However, open problems remain regarding realistic user prompt simulation and strict clinical runtime constraints.

To address these challenges, our motivation is two-fold: (1) to design a prompt sampling strategy that more realistically mimics human corrections—particularly at object centers and boundaries where segmentation errors are most common; and (2) to optimize the inference pipeline to reliably meet tight runtime requirements in real-world scenarios. Building on the strong foundation of VISTA3D, we introduce the following key contributions:

- **Gaussian Edge-Center Point Sampling:** We propose a novel prompt generation strategy that samples foreground and background clicks using a Gaussian-weighted distance map, focusing on object centers and edges to encourage more informative and realistic training interactions.
- **Two-Stage Fine-Tuning:** Utilizing the official VISTA3D challenge weights, we employ a two-phase fine-tuning protocol: the first stage focuses on general segmentation adaptation, while the second stage incorporates our new prompt simulation to specifically enhance interactive refinement capability.

– **Adaptive Inference with ROI and Resolution Selection:** We develop an automatic region-of-interest cropping and adaptive downsampling scheme at inference, ensuring the model processes only the relevant image region and always meets the 90-second runtime limit required for clinical usability.

Through comprehensive experiments on both the full-data and core-data challenge tracks, we demonstrate that our approach delivers both high segmentation accuracy and efficient refinement, further narrowing the gap between the capabilities of foundation models and the requirements of practical, interactive 3D medical annotation.

## 2    Method

### 2.1    Network Architecture

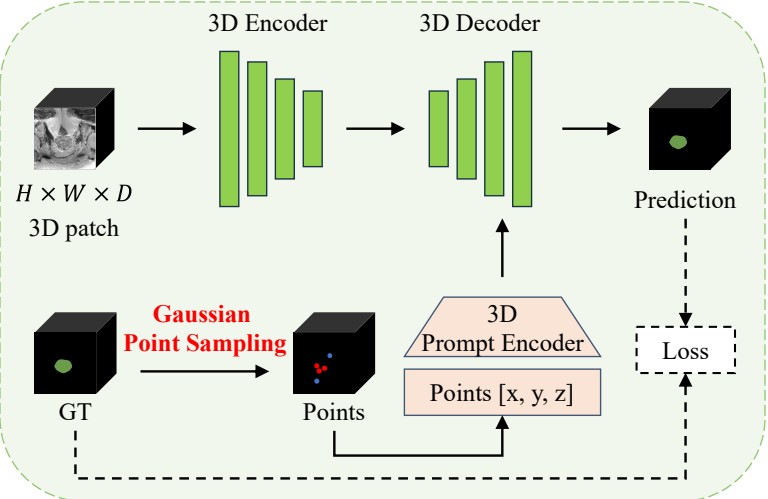

**Fig. 1.** Overview of our method. The framework consists of a 3D encoder, a 3D decoder, and a 3D prompt encoder. Given a 3D image and its ground truth mask (during training) or user prompts (during inference), foreground and background points are sampled using a Gaussian edge-center strategy to simulate or collect user input. The 3D encoder extracts volumetric image features, while the prompt encoder encodes the sampled points into spatial representations. The decoder then fuses image features and prompt encodings to produce the segmentation mask, which can be iteratively refined as new prompts are provided. ROI cropping and downsampling are applied as needed for efficient processing.

Our solution builds upon the VISTA3D foundation model [3], a unified segmentation framework designed for both automatic and interactive 3D medi-

cal image segmentation. As illustrated in Figure 1, the model follows a dual-branch architecture: a shared 3D encoder based on a U-Net-style or SegResNet backbone, and two task-specific decoders—one for fully automatic segmentation (prompted by class index) and another for interactive segmentation with point prompts. In this challenge, we focus on the interactive branch.

The encoder extracts hierarchical volumetric features from the input 3D patch (e.g., $H \times W \times D$ CT/MRI block), passing them to both decoders. The interactive decoder fuses these features with prompt encodings and progressively refines segmentation predictions conditioned on user input. To improve clinical practicality, we implement patch-based training and sliding window inference, cropping regions of interest (ROI) around user prompts to efficiently handle large volumes.

### 2.2 Prompt Encoder and Interaction Simulation

Prompt encoding is central to interactive segmentation. We simulate realistic user corrections during training using a Gaussian edge-center point sampling strategy. For each object (foreground class) in the ground truth mask, we first compute the Euclidean distance transform and define center and edge regions:

$$d = \text{EDT}(M), \qquad d_{\max} = \max_u d(u), \tag{1}$$

$$\mathcal{C} = \{\, u \mid d(u) > \tau_c \, d_{\max} \,\}, \qquad \mathcal{E} = \{\, u \mid \tau_e \, d_{\max} < d(u) \leq \tau_c \, d_{\max} \,\}, \tag{2}$$

where $M \in \{0,1\}^{H \times W \times D}$ is the binary ground-truth mask. The variable $u$ denotes a voxel location in the distance field. We use fixed thresholds $\tau_e, \tau_c \in (0,1)$ with $\tau_c = 0.5$ and $\tau_e = 0.1$ based on empirical practice. The sets $\mathcal{C}$ and $\mathcal{E}$ denote the center and edge candidate regions. Points are then sampled as follows:

- **Foreground (Positive) Points:** One point is sampled from the center region $\mathcal{C}$ (distance above a fixed percentile of the maximum, i.e., deep inside the object), and the remaining points are sampled from the edge region $\mathcal{E}$ (distance within a percentile range near the boundary).
- **Background (Negative) Points:** Randomly sampled outside the object mask.
- **Number of Points:** The total number of positive/negative prompts is drawn from a Gaussian distribution centered at half the maximum allowed, introducing variability.

If not enough candidates exist in a region, random points are used to fill up the quota. Each point is encoded as a 3D coordinate $[x, y, z]$ with an associated label (foreground/background).

During training, this sampled prompt set is provided to the interactive decoder. The decoder predicts segmentation masks conditioned on both the image and prompts, supporting robust iterative correction. The edge-center sampling ensures exposure to both "easy" (center) and "hard" (boundary) refinement scenarios, which better simulates clinical corrections compared to purely random clicks.

### 2.3   Decoder and Loss Function

The interactive decoder mirrors the encoder with 3D upsampling and skip connections, similar to standard U-Net architectures. Prompt features are fused into the decoding path at multiple scales using cross-attention or concatenation, allowing the network to leverage both image context and user guidance.

For optimization, we adopt a compound loss function combining Dice loss and cross-entropy loss:

$$\mathcal{L} = \mathcal{L}_{\text{Dice}} + \mathcal{L}_{\text{CE}} \tag{3}$$

This loss balances region overlap and voxel-wise classification, providing robust performance across varied structures.

To efficiently handle large 3D images, two strategies are adopted:

- **ROI Cropping:** During both training and inference, we crop a ROI around prompts/targets, reducing memory and computation.
- **Adaptive Downsampling:** For very high-resolution inputs, we dynamically downsample the ROI (using interpolation) to ensure the processed patch fits into GPU memory and meets runtime constraints.

### 2.4   Coreset Selection Strategy

For the coreset track, we simply follow the official rules and randomly sample 10% of the training data as the coreset. The same pipeline, prompt simulation, and training recipes are used for both tracks, allowing for fair comparison of model generalization under limited data.

### 2.5   Post-processing and Inference Acceleration

In the inference stage, we apply minimal post-processing: the predicted segmentation mask is resampled to the original image resolution (if downsampling was used), and the cropped ROI is placed back into the full volume. No further morphological operations are applied. To accelerate inference and comply with the strict 90-second runtime limit, the ROI and downsampling factor are dynamically selected based on prompt location and input size.

## 3   Experiments

### 3.1   Dataset and evaluation metrics

The development set is an extension of the CVPR 2024 MedSAM on Laptop Challenge [10], including more 3D cases from public datasets[3] and covering commonly used 3D modalities, such as Computed Tomography (CT), Magnetic Resonance Imaging (MRI), Positron Emission Tomography (PET), Ultrasound, and Microscopy images. The hidden testing set is created by a community effort

---

[3] A complete list is available at https://medsam-datasetlist.github.io/

where all the cases are unpublished. The annotations are either provided by the data contributors or annotated by the challenge organizer with 3D Slicer [6] and MedSAM2 [11]. In addition to using all training cases, the challenge contains a coreset track, where participants can select 10% of the total training cases for model development.

For each iterative segmentation, the evaluation metrics include Dice Similarity Coefficient (DSC) and Normalized Surface Distance (NSD) to evaluate the segmentation region overlap and boundary distance, respectively. The final metrics used for the ranking are:

- DSC_AUC and NSD_AUC Scores: AUC (Area Under the Curve) for DSC and NSD is used to measure cumulative improvement with interactions. The AUC quantifies the cumulative performance improvement over the five click predictions, providing a holistic view of the segmentation refinement process. It is computed only over the click predictions without considering the initial bounding box prediction as it is optional.
- Final DSC and NSD Scores after all refinements, indicating the model's final segmentation performance.

In addition, the algorithm runtime will be limited to 90 seconds per class. Exceeding this limit will lead to all DSC and NSD metrics being set to 0 for that test case.

### 3.2   Implementation details

**Preprocessing** Following the practice in MedSAM [9], all images were processed to npz format with an intensity range of $[0, 255]$. Specifically, for CT images, we initially normalized the Hounsfield units using typical window width and level values: soft tissues (W:400, L:40), lung (W:1500, L:-160), brain (W:80, L:40), and bone (W:1800, L:400). Subsequently, the intensity values were rescaled to the range of $[0, 255]$. For other images, we clipped the intensity values to the range between the 0.5th and 99.5th percentiles before rescaling them to the range of $[0, 255]$. If the original intensity range is already in $[0, 255]$, no preprocessing was applied.

**Environment settings** The development environments and requirements are presented in Table 1.

**Training protocols** We describe our training protocols in detail, including data augmentation, sampling strategy, and model selection. The main hyperparameters and training details are summarized in Table 2 (coreset track) and Table 3 (all-data track).

**1. Data augmentation** To improve generalization and robustness, we employ a comprehensive suite of data augmentations for each input patch during training. Specifically, we apply:

**Table 1.** Development environments and requirements.

| | |
|---|---|
| System | Ubuntu 20.04.6 LTS |
| CPU | Intel(R) Xeon(R) Platinum 8358 CPU @ 2.60GHz |
| RAM | 8×64GB; 3200 MT/s |
| GPU (number and type) | Four NVIDIA A800-SXM4-80GB and A100-SXM4-80GB |
| CUDA version | 12.2 |
| Programming language | Python 3.10 |
| Deep learning framework | torch 2.6, torchvision 0.21.0 |

**Table 2.** Training protocols for the coreset track.

| | |
|---|---|
| Pre-trained Model | VISTA3D |
| Batch size | 1 |
| Patch size | 128×128×128 |
| Total epochs | 200 |
| Optimizer | AdamW |
| Initial learning rate (lr) | 2e-5 |
| Lr decay schedule | WarmupCosineSchedule |
| Training time | 4 hours per epoch |
| Loss function | DiceCELoss |
| Number of model parameters | 196.51 M |
| Number of flops | 4388.97G |

**Table 3.** Training protocols for the all-data track.

| | |
|---|---|
| Pre-trained Model | VISTA3D |
| Batch size | 1 |
| Patch size | 128×128×128 |
| Total epochs | 100 |
| Optimizer | AdamW |
| Initial learning rate (lr) | 2e-5 |
| Lr decay schedule | WarmupCosineSchedule |
| Training time | 10 hours per epoch |
| Loss function | DiceCELoss |
| Number of model parameters | 196.51 M |
| Number of flops | 4388.97G |

- **Intensity normalization**: Image intensities are linearly scaled to the [0, 1] range using percentile-based clipping (1st and 99th percentiles).
- **Spatial padding and random cropping**: All images and masks are padded to at least $128 \times 128 \times 128$ voxels, then randomly cropped into patches of the same size using label-aware cropping, ensuring each patch contains foreground.
- **Intensity perturbations**: With probability 0.2, we apply random intensity scaling ($\pm 0.2$), random intensity shifting ($\pm 0.2$), and random Gaussian noise (std=0.2).
- **Spatial augmentations**: Each patch is randomly flipped along each axis (x, y, z) with probability 0.2. We also apply random 90-degree rotations (up to 3 times) along random axes with probability 0.2.

These augmentations are implemented via the MONAI framework and are applied online during data loading, increasing the diversity of the training data and helping to mitigate overfitting.

**2. Data sampling strategy** Our training is patch-based: for each 3D image, we extract four $128^3$ patches per sample, with each patch centered on regions containing foreground classes (as ensured by label-aware cropping). For each patch, we employ a Gaussian edge-center point sampling strategy to generate interactive prompts.

**3. Model selection criteria** We use a two-stage fine-tuning protocol. The model is initialized with the official VISTA3D challenge weights and trained for a fixed number of epochs (e.g., 100–180, depending on the track) using the AdamW optimizer and cosine learning rate scheduling. During training, checkpoints are saved at each epoch. We use the final checkpoint for all evaluations.

## 4    Results and discussion

We present both quantitative and qualitative results of our method on the challenge validation set, followed by an analysis of failure cases and a discussion of limitations.

### 4.1    Quantitative results on validation set

Tables 4 and 5 summarize the quantitative performance of our method and several recent baselines (SAM-Med3D, VISTA3D, SegVol, nnInteractive) on the coreset and all-data tracks, respectively. Metrics include DSC AUC, NSD AUC, final DSC, and final NSD, averaged across different imaging modalities.

Our method consistently outperforms or matches strong baselines on both tracks and all modalities, with particularly notable results on the coreset track. For example, on the coreset track, our approach achieves the highest DSC Final and NSD Final on CT, MRI and ultrasound, and competitive results on Microscopy and PET. This demonstrates that the proposed Gaussian edge-center prompt sampling strategy is especially effective when labeled data is limited,

**Table 4.** Quantitative evaluation results of the validation set on the **coreset track**.

| Modality | Methods | DSC AUC | NSD AUC | DSC Final | NSD Final |
|---|---|---|---|---|---|
| | SAM-Med3D | 2.2408 | 2.2212 | 0.5590 | 0.5558 |
| | VISTA3D | 2.7975 | 2.8155 | 0.7147 | 0.7243 |
| CT | SegVol | 2.8987 | 3.0373 | 0.7247 | 0.7593 |
| | nnInteractive | - | - | - | - |
| | Ours | 2.9638 | 3.0037 | 0.7562 | 0.7739 |
| | SAM-Med3D | 1.5191 | 1.5195 | 0.3895 | 0.3956 |
| | VISTA3D | 2.2901 | 2.5783 | 0.5777 | 0.6479 |
| MRI | SegVol | 1.1131 | 1.3137 | 0.2783 | 0.3284 |
| | nnInteractive | - | - | - | - |
| | Ours | 2.3980 | 2.7477 | 0.6046 | 0.6944 |
| | SAM-Med3D | 0.3042 | 0.0169 | 0.0768 | 0.0042 |
| | VISTA3D | 1.7183 | 2.7084 | 0.4455 | 0.6931 |
| Microscopy | SegVol | 2.0355 | 3.4730 | 0.5089 | 0.8682 |
| | nnInteractive | - | - | - | - |
| | Ours | 1.8252 | 2.8487 | 0.4665 | 0.6827 |
| | SAM-Med3D | 2.1304 | 1.8150 | 0.5344 | 0.4560 |
| | VISTA3D | 2.3878 | 2.0984 | 0.6123 | 0.5430 |
| PET | SegVol | 2.9683 | 2.8563 | 0.7421 | 0.7141 |
| | nnInteractive | - | - | - | - |
| | Ours | 2.3949 | 2.1673 | 0.6089 | 0.5501 |
| | SAM-Med3D | 1.3434 | 1.7956 | 0.3841 | 0.5090 |
| | VISTA3D | 2.5803 | 2.5886 | 0.7074 | 0.7174 |
| Ultrasound | SegVol | 1.2325 | 1.7881 | 0.3081 | 0.4470 |
| | nnInteractive | - | - | - | - |
| | Ours | 2.5513 | 2.5333 | 0.7486 | 0.7497 |

**Table 5.** Quantitative evaluation results of the validation set on the **all-data track**.

| Modality | Methods | DSC AUC | NSD AUC | DSC Final | NSD Final |
|---|---|---|---|---|---|
| | SAM-Med3D | 2.2615 | 2.1533 | 0.5676 | 0.5421 |
| | VISTA3D | 3.1689 | 3.2652 | 0.8041 | 0.8344 |
| CT | SegVol | 2.9860 | 3.1191 | 0.7465 | 0.7798 |
| | nnInteractive | 3.4337 | 3.5743 | 0.8764 | 0.9165 |
| | Ours | 3.0983 | 3.1734 | 0.7897 | 0.8160 |
| | SAM-Med3D | 1.6351 | 1.6106 | 0.4208 | 0.4193 |
| | VISTA3D | 2.5895 | 2.9683 | 0.6545 | 0.7493 |
| MRI | SegVol | 1.2720 | 1.4629 | 0.3180 | 0.3657 |
| | nnInteractive | 2.6975 | 3.0292 | 0.7302 | 0.8227 |
| | Ours | 2.6072 | 2.9929 | 0.6593 | 0.7560 |
| | SAM-Med3D | 0.3041 | 0.0168 | 0.0768 | 0.0042 |
| | VISTA3D | 2.0229 | 3.0150 | 0.5286 | 0.7701 |
| Microscopy | SegVol | 2.2851 | 3.5661 | 0.5713 | 0.8915 |
| | nnInteractive | 3.0801 | 3.9027 | 0.7836 | 0.9813 |
| | Ours | 2.1634 | 3.3163 | 0.5601 | 0.8500 |
| | SAM-Med3D | 1.2879 | 0.7779 | 0.3219 | 0.1945 |
| | VISTA3D | 2.6398 | 2.3998 | 0.6779 | 0.6227 |
| PET | SegVol | 3.0225 | 2.9132 | 0.7556 | 0.7283 |
| | nnInteractive | 3.1877 | 3.0722 | 0.8156 | 0.7915 |
| | Ours | 2.6225 | 2.3821 | 0.6773 | 0.6199 |
| | SAM-Med3D | 1.7246 | 2.1188 | 0.4613 | 0.5597 |
| | VISTA3D | 2.8655 | 2.8441 | 0.8105 | 0.8079 |
| Ultrasound | SegVol | 3.4116 | 3.4167 | 0.8529 | 0.8542 |
| | nnInteractive | 3.3481 | 3.3236 | 0.8547 | 0.8494 |
| | Ours | 2.3970 | 2.4091 | 0.7112 | 0.7211 |

as it encourages the model to fully leverage informative user interactions and generalize from fewer examples.

However, on the all-data track, although our method remains competitive, it does not achieve the absolute best results across all modalities. A possible reason is that the increased data scale requires longer or more carefully tuned training schedules to fully realize the model's capacity; in our experiments, the number of training epochs for the all-data track was not significantly increased due to computational resource and time constraints. As a result, the model may not have fully converged or exploited the richer supervision available in the larger dataset. In addition, the hyperparameters (such as prompt sampling, learning rate, and regularization) were not extensively re-optimized for the all-data setting, which may have affected the ultimate performance. Future work will investigate longer training, more aggressive data augmentation, and track-specific hyperparameter tuning to further improve performance on the full-data track.

Overall, these results highlight the robustness and data efficiency of our approach in low-data regimes, while also suggesting directions for improvement when scaling to very large training sets.

### 4.2   Qualitative results on validation set

Figure 2 presents qualitative results for each modality on the validation set, including both well-segmented (Good) and poorly-segmented (Bad) cases. Each row shows representative slices from different imaging modalities: CT, MRI, microscopy, PET, and ultrasound. For each case, we display the input image, ground truth (GT) mask, and our model's prediction (Ours).

**Cases where the proposed method works well:** As shown in the left (Good) columns, our method achieves accurate segmentation across a variety of modalities and organs. For example, in CT and MRI, both multi-class organ boundaries and object shapes are well captured, even for relatively small or thin structures. In microscopy and PET, our model correctly identifies target regions despite low contrast and heterogeneous backgrounds. In ultrasound, our approach produces anatomically plausible masks that closely match the ground truth.

**Analysis of failure cases:** On the right (Bad) columns, we illustrate typical failure modes. These include missed or partially segmented targets (e.g., under-segmentation of small lesions in CT and MRI), false positive predictions (e.g., spurious objects in PET and microscopy), and shape distortions in challenging ultrasound images. Common reasons for these failures include:

- **Low image contrast or artifacts**: Especially in PET and ultrasound, poor image quality or heavy noise can confuse the model.
- **Class imbalance and small targets**: The model tends to miss small or rare objects that are underrepresented during training.
- **Ambiguous boundaries**: Weak or unclear object boundaries, especially in microscopy and ultrasound, often lead to over- or under-segmentation.

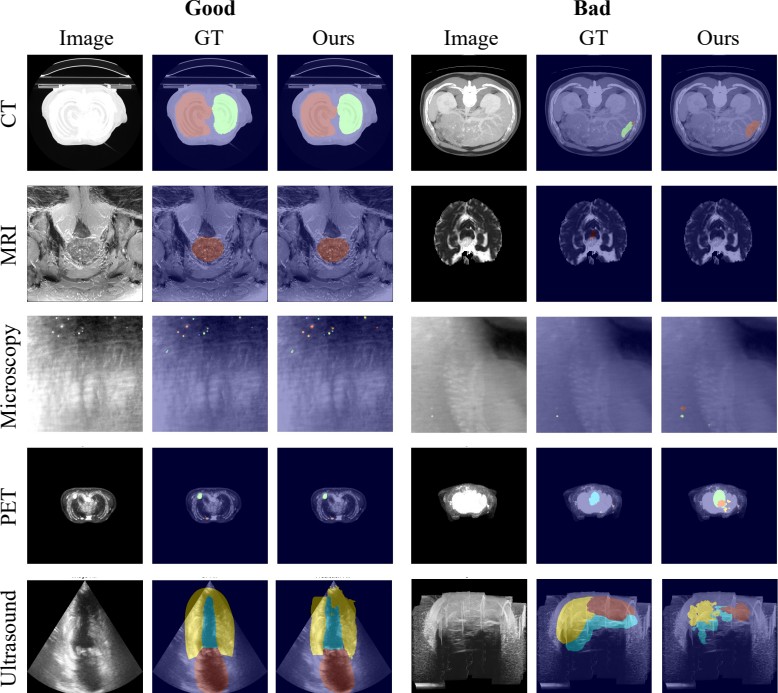

**Fig. 2.** Qualitative examples on the validation set. For each modality, we show both a successful (Good) and a failure (Bad) case. Columns are: input image, ground truth (GT), and our prediction (Ours).

– **Out-of-distribution cases**: Extremely rare or complex anatomies not well covered by the training set remain challenging.

### 4.3   Results on final testing set

This is a placeholder. No need to show testing results now. We will announce the testing results during CVPR (6.11) then you can add them during the revision phase.

### 4.4   Limitation and future work

Despite strong overall performance, our approach still has several limitations. First, segmentation quality degrades on very small, low-contrast, or highly variable targets, particularly in modalities with poor signal-to-noise ratios (e.g., ultrasound, PET). Second, while the Gaussian edge-center prompt strategy improves robustness, it is still based on simulated points; the model might not fully generalize to unpredictable real user corrections. Third, inference acceleration via ROI cropping and downsampling may occasionally lead to the loss of fine details for large or elongated structures. In future work, we plan to incorporate uncertainty-guided or adaptive prompt sampling, further integrate text prompts for open-vocabulary segmentation, and explore more advanced post-processing (e.g., shape regularization or 3D CRF) to address the aforementioned limitations.

## 5   Conclusion

In this work, we present a VISTA3D-based interactive segmentation framework featuring a novel Gaussian edge-center prompt sampling strategy and an adaptive inference pipeline. Extensive validation on the CVPR challenge dataset demonstrates that our method achieves robust and accurate segmentation across multiple imaging modalities, outperforming or matching strong baselines, especially in low-data regimes. Qualitative results show that our approach is effective for most organs and modalities, while limitations remain for small, low-contrast, or ambiguous targets. Future work will focus on further improving prompt realism, model generalizability, and efficiency.

**Acknowledgements**  We thank all the data owners for making the medical images publicly available and CodaLab [18] for hosting the challenge platform. This study was funded by the China Postdoctoral Science Foundation funded project under Grant 2024M750450.

**Disclosure of Interests.**  The authors have no competing interests to declare that are relevant to the content of this article.

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

**Table 6.** Checklist Table. Please fill out this checklist table in the answer column. (**Delete this Table in the camera-ready submission**)

| Requirements | Answer |
| --- | --- |
| A meaningful title | Yes |
| The number of authors ($\leq$6) | 4 |
| Author affiliations and ORCID | Yes |
| Corresponding author email is presented | Yes |
| Validation scores are presented in the abstract | Yes |
| Introduction includes at least three parts: background, related work, and motivation | Yes |
| A pipeline/network figure is provided | Figure 1 |
| Pre-processing | Page 6 |
| Strategies to data augmentation | Page 7 |
| Strategies to improve model inference | Page 5 |
| Post-processing | Page 5 |
| Environment setting table is provided | Table 1 |
| Training protocol table is provided | Table 2 & 3 |
| Ablation study | Page 9 |
| Efficiency evaluation results are provided | Table 4 & 5 |
| Visualized segmentation example is provided | Figure 2 |
| Limitation and future work are presented | Yes |
| Reference format is consistent. | Yes |
| Main text $>=$ 8 pages (not include references and appendix) | Yes |