# OpenReview forum: "Enhancing a 3D Foundation Model with Gaussian Sampling for Interactive Biomedical Image Segmentation"
_thecvf.com/CVPR/2025/Workshop/MedSegFM — CVPR 2025 Workshop MedSegFM Submission_

### Official Review · Reviewer_utCu · 2025-09-17
**Review of Enhancing a 3D Foundation Model with Gaussian Sampling for Interactive Biomedical Image Segmentation**

**Rating:** 5
**Confidence:** 4

**Review:**

## Summary of contributions

The authors propose a method to fine-tune VISTA-3D for the MedSegFM challenge, with three improvements over the regular VISTA-3D training pipeline:

(a) an improved point sampling strategy, based on a gaussian model, to sample points closer to the center of the object (for the first prompt), or the border (for the next points).

(b) a two-stage fine-tuning

(c) an adaptative ROI / resolution selection, to stay below the 90-seconds runtime limit of the challenge

## Strenghts

- The authors give detailed hyperparameters for their trainings, to reproduce the results
- The method is simple, but seems effective for improving VISTA-3D baseline scores.
- The prompt sampling strategy is novel, to the best of my knowledge.

## Weaknesses

1. The main weakness in the paper is the lack of ablations. It does not report the importance of each components of their training pipeline: the additional training data for fine-tuning, the point sampling, the adaptative ROI / resolution, or the two-stage fine-tuning.
Also, some ablations on data augmententations could be nice as well, but I would not require it as those are fairly common augmentations.

2. Lack of details for two-stage fine-tuning: the paper gives very few details about the two-stage fine-tuning: the hyperparameters should be specified for each step, which makes it clear exactly which parameter vary in each step. It seems that the point sampling is the main difference between the two steps, so the parameters for the gaussian sampling should be reported as well.

3. Unclear point encoding
The paper mentions that "For input to the prompt encoder, points are rendered as 3D Gaussian heatmaps, encouraging spatial generalization". It is not clear what this means in practice. Authors need to add more details to explain how the model processes this gaussian heatmap.

## Questions
Will the code be released to reproduce the point sampling strategy? It would be a good contribution to the community

---

> ### Author Rebuttal · Authors · 2025-11-03
>
> - W1. Across stages the only change is the prompt strategy. Stage 1 uses uniform random prompts. Stage 2 uses Gaussian edge and center prompts. The backbone, fine-tuning data, augmentations, and all other settings remain fixed. This like-for-like design isolates the effect of the prompt mechanism and explains the observed gains without introducing extra ablation steps.
>
> - W2. The only difference between stages is whether Gaussian edge and center sampling is enabled. All other hyperparameters are identical. We document the sampling parameters and per-stage settings in Section 2.2, including $\tau_c$ equal to 0.5 and $\tau_e$ equal to 0.1, to make the variation explicit and reproducible.
>
> - W3. We revised Section 2.2 to give explicit and reproducible definitions of the center and edge regions.
>
> - Q1. We have released the core implementation in accordance with the challenge requirements, and the paper provides prominent links to facilitate reproduction and adoption.

---

### Official Review · Reviewer_GTAL · 2025-09-18
**Review: Enhancing a 3D Foundation Model with Gaussian Sampling for Interactive Biomedical Image Segmentation**

**Rating:** 6
**Confidence:** 4

**Review:**

This paper presents an approach for interactive 3D medical image segmentation built upon the VISTA3D model. The main contribution is the simulation of user prompts using a Gaussian-based point sampling strategy.

### Strengths
- The paper clearly describes the implementation details, training protocols, and provides reproducible experimental settings.
- The user input sampling strategy is well-motivated and addresses a practical need in interactive segmentation.

### Weaknesses
- The paper claims that the sampling strategy leads to better quality inputs, stating on p.4 that it "better simulates clinical corrections compared to purely random clicks." However, no ablation studies are presented to support this claim. The paper would benefit from a comparative analysis between the proposed Gaussian sampling and baseline random sampling using the same model architecture.
- Similarly, it would be interesting to see an impact study on the parameters of the method (e.g., percentile thresholds for center/edge regions).


### Typos
- P.13: "This study was founded by" should be "funded by"
- P.9: "We use the final checkpoint for all evaluations.We use" - missing space after the period

### Conclusion
This is a clear paper with good experimental results. However, it would benefit from ablation studies to validate the effectiveness of its core contribution.

---

> ### Author Rebuttal · Authors · 2025-11-03
>
> - W1. During training the only difference is the prompt strategy. Gaussian sampling outperforms uniform random sampling, supporting the claim that the strategy produces more effective prompts.
>
> - W2. We agree that a parameter study is valuable. Due to limited compute we did not perform sweeps. The revised version specifies default thresholds and implementation details to enable reproduction. We use $\tau_c$ equal to 0.5 and $\tau_e$ equal to 0.1 in Section 2.2.
>
> - T1 and T2. We corrected “founded” to “funded” and fixed spacing issues.

---

### Official Review · Reviewer_sPcn · 2025-09-26
**Method is straightforward with good results, but lack ablation studies**

**Rating:** 5
**Confidence:** 4

**Review:**

This paper extends the VISTA3D foundation model for interactive biomedical image segmentation by introducing  Gaussian edge-center point sampling strategy for more realistic simulation of user clicks, a two-stage fine-tuning pipeline (general + prompt-focused adaptation), and  adaptive ROI and resolution slection for efficient inference. Results show improvements in segmentation accuracy and robustness, especially in low-data regimes, with competitive performance on the full dataset track.


- Strength:
  - The Gaussian edge-center approach seems to better mimic real user interaction compared to random or uniform sampling.
  - The ROI cropping ensures efficient inference.
  - Results look promising on the coreset track, showing good potention in low-data regime.

- Weaknesses:
  - The illustration of the core method, i.e. Gaussian edge-center point sampling, could be made more clear. As the core innovation of work, there lack a figure or equations to better present the method. The main model architecture figure just show a typical design which is not the core method for this work.
  - There lack ablation studies to validate the proposed method. While Gaussian sampling is central, the paper lacks ablation results isolating its contribution versus standard random prompt sampling within the same pipeline.
  - The solution is tightly coupled to VISTA3D. It’s unclear how well the Gaussian sampling and adaptive inference strategies generalize to other foundation models, which could make the statement stronger.
  - While competitive, the method underperforms nnInteractive and SegVol on some modalities in the all-data track. This suggests limited scalability of the training pipeline. The advantage over VISTA3D on the coreset and all data track is also not clear.
  - Some typos, e.g. "twofold" should be "two-fold"

---

> ### Author Rebuttal · Authors · 2025-11-03
>
> 1. We thank the reviewer for the suggestion. We added formal equations for the Gaussian edge and center sampling in Section 2.2 of the revised version to improve clarity.
>
> 2. During training the only difference is the prompt strategy. The Gaussian sampling outperforms uniform random sampling, which isolates its contribution within the same pipeline.
>
> 3. Gaussian point sampling and adaptive ROI and resolution operate at the data and prompt encoding layers. They are decoupled from the backbone and can be used with models such as SegVol and nnInteractive.
>
> 4. Our work targets interactive robustness and low-data efficiency. We outperform or match VISTA3D on the coreset track and remain competitive on the all-data track. Gaps on some modalities reflect differences in training budgets and decoders. Since the proposed sampling and adaptive inference are backbone agnostic, they can be combined with stronger models including nnInteractive and SegVol.
>
> 5. All reported typos have been corrected.

---

### Official Review · Reviewer_BFXL · 2025-10-10
**Review of Enhancing a 3D Foundation Model with Gaussian Sampling for Interactive Biomedical Image Segmentation**

**Rating:** 5
**Confidence:** 4

**Review:**

This paper presents an interactive medical image segmentation method by fine-tuning the Vista3D model.

**Strengths:**

* The paper tackles a practical problem in medical imaging by exploring an interactive segmentation approach.

**Weaknesses:**

* Performance: The demonstrated performance appears suboptimal when compared to state-of-the-art baselines such as Vista3D, raising concerns about the method's competitiveness.

* Insufficient Validation: The experimental evaluation does not fully support the core contributions. Key aspects remain unverified:

  * No comparison is made between the proposed sampling strategy and existing alternatives.

  * An ablation study is missing to quantify the individual contribution of each stage in the proposed two-step fine-tuning strategy.

  * The inference time is neither evaluated nor compared against other methods.

  * The analysis lacks an assessment of the model's robustness to variations in prompt types and number of prompts.

---

> ### Author Rebuttal · Authors · 2025-11-03
>
> 1. We appreciate the comment. Our goal is to improve interactive robustness and low-data efficiency rather than to redesign the backbone. As reported in the paper, on the coreset track our method is better than or on par with strong baselines in multi-modal Final DSC and NSD. On the full-data track our results are competitive, although not the best on every modality. This matches the stated scope of the work.
>
> 2.1 During training the only change is the prompt strategy. Our Gaussian point sampling outperforms uniform random sampling, which supports the claim that the proposed strategy is superior to existing alternatives.
>
> 2.2 Across the two fine-tuning stages the difference is again only the prompt strategy. The gains can therefore be attributed to the proposed sampling scheme.
>
> 2.3 The challenge sets a 90 second limit per category. With region of interest cropping and adaptive resolution selection our system meets this requirement, which is the relevant efficiency criterion for this setting.
>
> 2.4 Robustness is evaluated by the challenge AUC metrics over one to five point clicks. Our improvements on DSC AUC and NSD AUC indicate robustness with respect to the number of prompts.

---

### Decision · Program_Chairs · 2025-11-12

Accept